# Modelling Cancer Metastasis in *Drosophila melanogaster*

**DOI:** 10.3390/cells12050677

**Published:** 2023-02-21

**Authors:** Joanne L. Sharpe, Jason Morgan, Nicholas Nisbet, Kyra Campbell, Andreu Casali

**Affiliations:** 1School of Biosciences, The University of Sheffield, Sheffield S10 2TN, UK; 2Departament de Ciències Mèdiques Bàsiques, Universitat de Lleida and IRBLleida, Av. Alcalde Rovira Roure, 80, 25198 Lleida, Spain

**Keywords:** Drosophila, cancer, metastasis, larva, adult

## Abstract

Cancer metastasis, the process by which tumour cells spread throughout the body and form secondary tumours at distant sites, is the leading cause of cancer-related deaths. The metastatic cascade is a highly complex process encompassing initial dissemination from the primary tumour, travel through the blood stream or lymphatic system, and the colonisation of distant organs. However, the factors enabling cells to survive this stressful process and adapt to new microenvironments are not fully characterised. Drosophila have proven a powerful system in which to study this process, despite important caveats such as their open circulatory system and lack of adaptive immune system. Historically, larvae have been used to model cancer due to the presence of pools of proliferating cells in which tumours can be induced, and transplanting these larval tumours into adult hosts has enabled tumour growth to be monitored over longer periods. More recently, thanks largely to the discovery that there are stem cells in the adult midgut, adult models have been developed. We focus this review on the development of different Drosophila models of metastasis and how they have contributed to our understanding of important factors determining metastatic potential, including signalling pathways, the immune system and the microenvironment.

## 1. Introduction

*Drosophila melanogaster* (*Drosophila*) have been widely used to study the molecular and genetic underpinnings of human cancer [1]. Historically, *Drosophila* research has helped to identify the mechanisms of action of many pathways that play a key role in cancer, including BMP, Hedgehog, Hippo, JAK/STAT, Notch, Ras, TGFβ and Wnt. The development of new techniques such as MARCM [2], which makes it possible to generate homozygous clones of mutations in an animal that is otherwise heterozygous for the same mutation [3], enabled the generation of new models that mimic the loss of heterozygosity observed in the somatic cells of cancer patients. Since then, many multi-hit models of cancer have been described and used to study different aspects of the disease, for instance, the association between cancer and obesity [4,5], tumour–host interactions [6,7], genomic instability [8], inflammation and immunity [9] and cancer cachexia [10]. Since many aspects of *Drosophila* as a model for cancer and human diseases have been extensively and nicely reviewed elsewhere [1,11,12], we will focus this review on the use of *Drosophila* as a model for cancer metastasis.

## 2. Cancer Metastasis

Tumour metastasis is a complex multistage process during which malignant cells spread from a primary tumour and proliferate, forming secondary tumours at distant sites (Figure 1). From the cells that are released from a tumour, only a small proportion form a distant secondary tumour. This is because very few cells are able to accumulate the phenotypic traits, akin to stem cells or regenerative stem cells, that are required to survive the stress related to the processes of cell dissemination, adaptation to a distant niche and grow [13,14,15,16,17].

Metastasis starts when tumour cells are able to leave the primary tumour and disseminate to distant organs (Figure 1). A key event in promoting this initial step is the transition of epithelial tumour cells towards a more mesenchymal cell state through an epithelial-to-mesenchymal transition (EMT), with the consequent disruption of cellular adhesions, loss of apical–basal polarity and drastic remodelling of the cytoskeleton. The acquisition of mesenchymal characteristics increases migratory capacity, invasiveness and resistance to apoptosis [18]. These changes allow tumour cells to migrate through the extracellular matrix and enter blood vessels, becoming circulating tumour cells (CTCs). CTCs may travel alone or in clusters [19] before becoming trapped in the capillaries of distant organs seconds to minutes after leaving the primary tumour. From there, they pass through the capillary endothelium to the parenchyma of organs (Figure 1). Alternatively, metastatic cells may disseminate through lymphatic vessels and, in some tumours, through routes that do not require entry into the circulation [20,21,22]. Once tumour cells have infiltrated distant organs, many are eliminated by lack of an appropriate microenvironment, together with the defensive activities of resident immune cells. However, a few malignant cells may enter a proliferative quiescence, known as the dormancy phase, that protects them from being eliminated. These cells may remain dormant for years, likely controlled by a balance between mitogenic and anti-mitogenic signals; when this balance is broken, the cells enter into the colonisation phase, outgrowing and forming an overt metastatic secondary tumour (Figure 1) [18,23].

Each phase of metastasis reflects the capacity of metastatic cells to evade immunity and to dynamically adapt to new microenvironments through a high phenotypic plasticity. However, the factors driving the dynamic cellular transitions of tumour cells that allow them progress through the different stages of metastasis are largely unknown. Moreover, genomic and transcriptomic studies have led to new insights into the intratumour heterogeneity of primary tumours and how this increases as metastatic cells evolve under the pressure of somatic mutations and clonal selection [24,25,26,27,28,29].

To better understand the complex processes driving metastasis, which currently is the leading cause of cancer-related deaths, research heavily relies on in vivo experimental models. The most widely used organism to model metastasis is mice, in which transplantation experiments and genetically engineered mouse models have provided very useful insights [30]. However, despite important caveats such as an open circulatory system and the lack of adaptative immunity, non-mammalian model organisms such as *Drosophila* have also proven very useful to understand the complex choreography of gene expression driving cell plasticity of tumour cells and their adaptation to new microenvironments, thanks to their amenability to complex genetic manipulations and experimental tractability. Here we will discuss the different ways *Drosophila* have been used to either study distinct stages of metastasis, or the entire process from primary tumour initiation to growth of secondary metastasis.

## 3. Modelling Metastasis in *Drosophila* Larvae

Generally, tumour cells arise from mutations in cells that undergo mitosis. Therefore, the capacity for neoplastic transformation depends primarily on the ability of the cells to divide. This makes the larval stages of *Drosophila* development fruitful ground for modelling cancer, as a number of cells and tissues undergo large bursts of proliferation: the imaginal disc cells; the adult optic neuroblasts and ganglion cells in the larval brain; the blood cells; the cells in the gonads; and other smaller cell nests within different organs (Figure 2A) [31]. It is, therefore, not so surprising that the first invasive tumours were discovered arising from the larval imaginal discs—these resulted from mutagenesis screens in the 1930s—which led to the discovery of the first *Drosophila* cancer genes [31].

The epithelial imaginal discs subsequently spawned a wealth of studies on the genetic control of epithelial organisation and its relation to invasive outgrowth. Around 80% of cancers are derived from epithelial tissues, and loss of tissue integrity features prominently in the progression of an epithelial tumour from benign to metastatic [36,37]. Owing to their accessibility and genetic tractability, the imaginal discs have become popular models for studying how changes to epithelial architecture link to the initiation of metastatic dissemination.

The first cancer genes discovered in *Drosophila* were found to organise epithelial polarity and differentiation [38]. Notably, larvae with recessive mutations in *Lethal Giant Larvae* (*lgl*)—a polarity regulator—develop sizeable neuroblastomas in the optic centre of the midbrain with evident invasion into the neuropile and an observed two-fold enlargement of the brain hemispheres [31]. *lgl* mutant neuroblasts of the eye disc fail to differentiate into ganglion mother cells (GMCs) and therefore do not enter a post-mitotic state, resulting in excess proliferation. Similar outcomes can be observed following recessive mutations to the polarity regulator *Discs Large* (*dlg*), as well as *Brain Tumour* (*brat*), and *Malignant Brain Tumour* (*mbt*), which are both GMC differentiation determinants [39]. These models recapitulate the loss of polarity, cell adhesion and resultant failure to segregate differentiation determinants that strongly correlate with metastatic progression in human epithelial tumours. Although *lgl*, *dlg*, *brat* and *mbt* mutations can invoke invasion from the eye disc into surrounding neural tissue, none have been observed to drive colonisation of distant organs within larvae [38,40,41]. Their metastatic capacity was later demonstrated by transplant assays—which we will discuss later.

Focusing more specifically on metastatic dissemination, Pagliarini et al. designed a genetic screen in larvae to identify mutations in genes that enable tumour cells of the eye imaginal disc to colonise distant sites [39]. Upregulation of *Ras^V12^* under the eye-disc-specific promoter *Eyeless* causes the formation of non-invasive tumours. *Ras^V12^* was overexpressed in clones of eye imaginal disc cells in combination with recessive mutations in candidate genes to identify mutations that cooperate with constitutively activated Ras to drive colonisation to distant tissues. Using this approach, they found that the cooperation of *Ras^V12^* with the loss of the polarity factor *scribble* (*scrib*) (*RasV12*; *scrib*) was sufficient to cause degradation of the basement membrane, transcriptional downregulation of the E-Cadherin gene *shotgun* and invasion into the ventral nerve chord (VNC) and haemolymph, as well as the formation of secondary foci at distant tissues (Figure 2B). The combination of *Ras^V12^* with mutations in the polarity regulators *lgl*, *dlg*, *stardust (sdt)*, *bazooka (baz)* and *cdc42* also produced similar metastatic behaviours. Importantly, although combining *Ras^V12^* with mutations in genes required for apicobasal polarity led to metastasis, mutations in genes required for apicobasal polarity alone resulted in loss of polarity and tumour outgrowth but no metastasis, as cells underwent apoptosis [39].

Collectively, these findings support a mechanism whereby loss of tissue architecture—particularly as it relates to a loss of epithelial polarity and differentiation— is a core attribute acquired by cancer cells in realising their metastatic potential. Nonetheless, it is insufficient to drive the process as epithelia appear to recognise that their integrity is compromised and undergo programmed cell death, potentially as an inbuilt tumour suppressor mechanism to prevent the outgrowth of malignant cells. The fact that the addition of *Ras^V12^* can overcome this has set the stage for a series of subsequent investigations into how oncogenes and tumour suppressor genes conspire together to enable metastatic behaviours that they otherwise would not be capable of executing alone.

It was later discovered that loss of polarity in *scrib* clones of the eye disc results in apoptosis through c-Jun N-terminal kinase (JNK)-mediated stress signalling [42]. Suppressing JNK-mediated apoptosis by expressing the anti-apoptotic baculovirus protein P35 was capable of enabling the metastasis of eye disc tumours [43]. Paradoxically, JNK signalling has also been found to be responsible for the invasive features in *scrib* clones. Indeed, the expression of a dominant negative, non-activatable form of JNK is capable of preventing metastases in *Ras^V12^*; *scrib* mutant imaginal discs [44]. A later study discovered that JNK acts mechanistically by upregulating Matrix Metalloproteinase 1 (MMP1) through the transcriptional action of *Drosophila* Fos (*dFos* or *Kayak*). When *Mmp1* is silenced by RNAi or antagonised with Tissue Inhibitor of Metalloproteases (TIMP), no basement membrane degradation or subsequent metastasis is observed [45]. These findings suggest that *Ras^V12^* contributes to metastasis by defusing the apoptotic circuitry of JNK, while leaving its invasive programming untampered. The same metastatic phenotype can also be produced by substituting *Ras^V12^* with the oncogene *Notch* [42]. Together, these findings suggest that metastasis from larval imaginal discs depends on the delicate collaboration of a tumour suppressor gene— which enables the overhaul of tissue integrity— as well as the input of an oncogene, which disarms apoptotic suppression by JNK.

Besides the intraclonal cooperation of cancer genes, similar metastatic outcomes can be produced when those genes are distributed among separate clones in the same tissue (Figure 2B). The ability to generate multiple genetic mosaics in the imaginal discs enabled investigation of the interclonal cooperation of clones carrying separate cancer genes. In one study, *Ras^V12^* clones were generated in the eye imaginal disc alongside adjacent clones with *scrib* mutations (*Ras^V12^/scrib*). These clones showed the same mechanisms of growth as *Ras^V12^*; *scrib* and result in similar metastatic outcomes, with visible invasion observed in the VNC [32]. Later studies demonstrated that such cooperation can enable invasion in a way that is not possible by a single clone sharing both genes. For example, Enomoto et al. demonstrated that single eye disc clones harbouring mutations in the oncogenes *Src* and *Ras* do not undergo invasion. However, by inducing separate clones expressing *Src* and *Ras*, it was found that both populations of clones invade into the VNC [46]. *Src* clones were seen to express the Notch receptor, whereas *Ras^V12^* clones were observed to express its ligand, Delta. The interaction of Notch with its cognate ligand triggered the downregulation of *E-Cadherin* expression in both clones, as well the additional downregulation of the pro-apoptotic factor head involution defective *(hid)* in Src+ cells. Interestingly, RNAi interference against *E-Cadherin* was insufficient to phenocopy invasion of *Src* clones into the VNC. This only occurred following the transcriptional silencing of both *E-Cadherin* and *hid*. This, again, suggests that larval tumours rely on the combined loss of tissue architecture and resistance to apoptosis.

Ohshawa et al. have shown that simultaneous intraclonal and interclonal cooperation can be required for invasive behaviours [34]. Following genetic screening, a number of genes encoding mitochondrial respiration complexes were found to collaborate intraclonally with *Ras^V12^* to induce the non-cell autonomous growth of surrounding *Ras^V12^* clones. The intraclonal combination of *Ras^V12^* and mitochondrial mutations (*Ras^V12^; mito −/−*) collaborating interclonally with *Ras^V12^* clones was found to drive invasion into the VNC and brain hemispheres. Mechanistically, increased superoxide levels resulted in the JNK-dependent secretion of Unpaired (Upd) and Wg cytokines. These cytokines were necessary for invasion, as abrogating Upd signalling prevented invasion from occurring. This collaboration of diverse populations of cancer clones is relevant as cancers host highly heterogenous populations of cells carrying variable mutations; such polyclonal modelling is likely to more accurately reflect how metastasis actually occurs.

Besides cancer cells collaborating with one another, there is also evidence that cancer cells actively compete with cells that are not one of their own (Figure 2C). The genetic tractability of *Drosophila* make them particularly amenable to modelling this phenomenon of “cell competition”, which refers to how cells within a heterogenous population will work against each other to attain a growth advantage, typically by securing a monopoly on growth factors in their niche [47,48,49]. In a recent study, Eichenlaub et al. overexpressed EGFR and *mIR-8* in the cells of the wing imaginal disc. This resulted in development of aggressive neoplasms that colonized distant tissues, including the hindgut [35]. A subset of these cells developed into “giant cells”, which had a considerable increase in overall cell size, enlarged polyploid nuclei and delocalisation of E-Cadherin and Dlg. Peculiarly, the subset of giant cells was flanked by differentiated wild-type cells with apoptotic signifiers such as pyknotic nuclei and cleaved-caspase 3 expression. The smaller wild-type cells were observed to be engulfed by the giant cells in a process that is dependent on the induction of apoptosis, as treatment with the apoptosis inhibitors P35 and DIAP1 prevented the formation of giant cells. Importantly, this same treatment also inhibited colonisation, suggesting that, in this context, metastasis proceeds by cell competition involving apoptosis of neighbouring wild-type cells [35].

## 4. Using *Drosophila* Larvae to Study Interactions with the Microenvironment

Since the first *Drosophila* cancer genes were identified in larvae in 1930, many have capitalised on the genetic tractability and suitability of this system to unravel key pathways implicated in tumorigenesis and metastasis. More recently, this system has been utilised to examine the effect of manipulating the microenvironment on tumour growth and malignancy.

It was famously proposed that tumours resemble wounds that do not heal [50]. Indeed, many of the cellular and molecular alterations recurrent in wound healing become reactivated in cancer, often to the benefit of metastasis. This response is largely due to the microenvironment, consisting of stromal cells, signalling molecules, blood vessels and the extracellular matrix (ECM) (Figure 3A) [51]. Although cancers harbour highly heterogenous populations of clones, which both cooperate and conspire to eliminate their neighbours, stromal cells can account for as much as 90% of a tumour mass [52,53]. In addition, immune cells contribute to metastasis through the secretion of cytokines, which heavily influence metastatic spread via chemotaxis [54]. Furthermore, systemic inflammation is one of the central means by which metastasis is so characteristically lethal [55]. Although *Drosophila* lack adaptive immunity and possess an open circulatory system, studies on larvae have nonetheless demonstrated a remarkable degree of conservation as regards to the interplay between metastasis and the microenvironment (Figure 3A).

In *Drosophila* larval epithelia, a wound will activate the JNK and the Janus kinase (JAK)/signal transducer and activator of transcription (STAT) pathway stress signalling circuitry, which represents a strong component of Drosophila innate immunity [56]. JNK activation leads to degradation of the basement membrane in a MMP1-dependent manner. This results in the delamination of old and injured cells, which are quickly cleared through JNK’s apoptotic machinery response [57]. Moreover, JNK will drive the secretion of inflammatory cytokines through JAK/STAT signalling, which in turn acts to recruit haemocytes to the wound, as well as coaxing the fat body and remote haemocytes to secrete their own ligands in a systemic inflammatory response (Figure 3A) [57].

Benign imaginal disc tumours driven by mutations in the polarity factors *dlg*, *lgl* and *scrib* appear to activate a similar inflammatory response, with recruited haemocytes secreting Eiger (*Drosophila* TNFα), which triggers tumour cell apoptosis through JNK signalling [58]. When *eiger* (*egr*) is experimentally downregulated, these tumours grow significantly (Figure 3D). By contrast, when *egr* is equivalently downregulated in the metastatic *Ras^V12^*; *scrib* model, invasion is prevented and tumour growth is reduced [59] (Figure 3D). This is likely to relate to the apoptotic insensitivity conferred by *Ras^V12^*, which leaves the cells unscathed and still capable of harnessing the invasive capabilities afforded to them by JNK signalling.

In another study, Mishra-Gorur et al. demonstrated that *Ras^V12^; scrib* tumours exhibit preferential colonisation—that is, organotropic metastasis—of the VNC and the mouth hooks but not other sites such as the wing disc. A genome-wide RNAi screen established that silencing the *Toll-6* receptor or its ligand *Spätzle*, not only abolishes organotropic metastasis but inhibits invasion entirely (Figure 3E) [60]. Mechanistically, Spätzle was found to be secreted by the metastasis-receptive sites and is evidenced to engage JNK signalling by binding Toll-6 receptors on the imaginal disc cells. As it has already been demonstrated that *scrib* clones are subject to immune surveillance [58]; the presentation of Toll-6 likely represents a second means by which these cells make themselves amenable to immune destruction, with apoptotic resistance through *Ras^V12^* representing an avenue by which they evade surveillance. Together, these findings suggest that, as in vertebrate systems, larval tumours resemble wounding environments that, at the expense of benign tumours, become reprogrammed to their advantage as they assume malignancy.

Tumour cells must migrate considerable distances to metastasise to distant sites. Blood vessel remodelling has been the subject of intense research, partially by virtue of its capacity to disseminate cancer cells to remote organs [61]. Although flies lack blood vessels, a number of studies have identified a synonymous phenomenon in the form of neo-tracheogenesis. *Ras^V12^*; *scrib* eye disc tumours have been observed to recruit and invade tracheal tubules [39]. In other studies, cancer cells have been documented to crawl along the surface of tubules, even over considerable distances [62]. Indeed, these findings extend beyond local invasion. Calleja et al. found that *lgl* mutant wing, leg and haltere disc tumours depend on tracheal tubule remodelling to colonise the CNS [63]. These *lgl* cells were found to suffer hypoxia, as indicated by the hypoxia-specific lactate dehydrogenase reporter, and subsequently recruit tracheal tubules through the secretion of Branchless (Bnl), a fly homolog of the common angiogenesis factor Fibroblast Growth Factor (FGF) (Figure 3C). Moreover, some *lgl* cells were noted to transdifferentiate into pseudo-tracheal cells expressing MMP1, which colocalised with basement membrane breaks. This may represent a means by which imaginal disc tumours undergo invasion. It also resembles human cancers, where some tumour cells are documented to transdifferentiate to endothelial precursors giving rise to blood vessels [64].

Grifoni et al. have provided some insights into the mechanisms underlying neo-tracheogenesis in cancer models: in *Ras^V12^*; *lgl* wing disc hypoxia is sensed by the Hypoxia-inducible factor 1-alpha (HIF1A) orthologue Similar (Sima), which upregulates Bnl to attract tracheal tubules through their FGF receptor, Breathless (Btl) [65]. Moreover, cancer cells exhibit directional movement, with extended filopodia in the direction of recruited tracheal tubules, indicating that they are invading out in a coordinated fashion under chemotactic cues. Further examination will be needed to unpack how strongly tracheal remodelling contributes to the invasion and transport of disseminating tumour cells.

A surge in pro-inflammatory cytokines driving systemic inflammation is associated with an increase in metastasis [66]. Notably, a number of cytokines such as Tumour necrosis factor (TNFα), Transforming growth factor β (TGFβ), and Interleukin 6 (IL-6) have been tied to the progressive wasting of muscle and adipose tissue through a process termed cachexia [67]. Emerging evidence suggests that cytokines activate autophagy in these tissues to recycle nutrients for the growing metabolic demands of invasive tumours [68]. Cachexia is estimated to be responsible for around 20–40% of cancer deaths and is a central underlier of metastasis-associated mortality [69]. Cachexia has been observed in larval tumour models, allowing investigation into links with tumour invasion and metastasis. One study on the transformed eye disc found that metabolically stressed cells in the metastatic *Ras^V12^*; *scrib* model, but not the *Ras^V12^* benign model, secrete Upd1-3 cytokines that cause systemic autophagy in the muscle, fat body and midgut [70]. Pharmacological or genetic ablation of autophagy using chloroquine or *Autophagy-related protein 13* (*Atg13*) knockout, respectively, resulted in significantly reduced invasion into the VNC [70] (Figure 3B). In a later study, evaluating muscle and adipose tissue volume using Computed-Tomography (CT), it was found that *Ras^V12^*; *scrib* tumours grow 10-fold when invading into the VNC, while seeing a 50% reduction in muscle volume compared with the benign *Ras^V12^* control [71]. Moreover, liquid chromatography/mass spectrometry detected an increase in circulating sugars and amino acids as autophagy proceeds, with carbon-13 tracing showing that the tumours become progressively more reliant on deteriorating tissues for nutrition as they grow [71]. These findings draw attention to the potential of autophagy as a target for pharmacological intervention in the prevention of metastasis-associated mortality and morbidity.

**Figure 3 cells-12-00677-f003:**
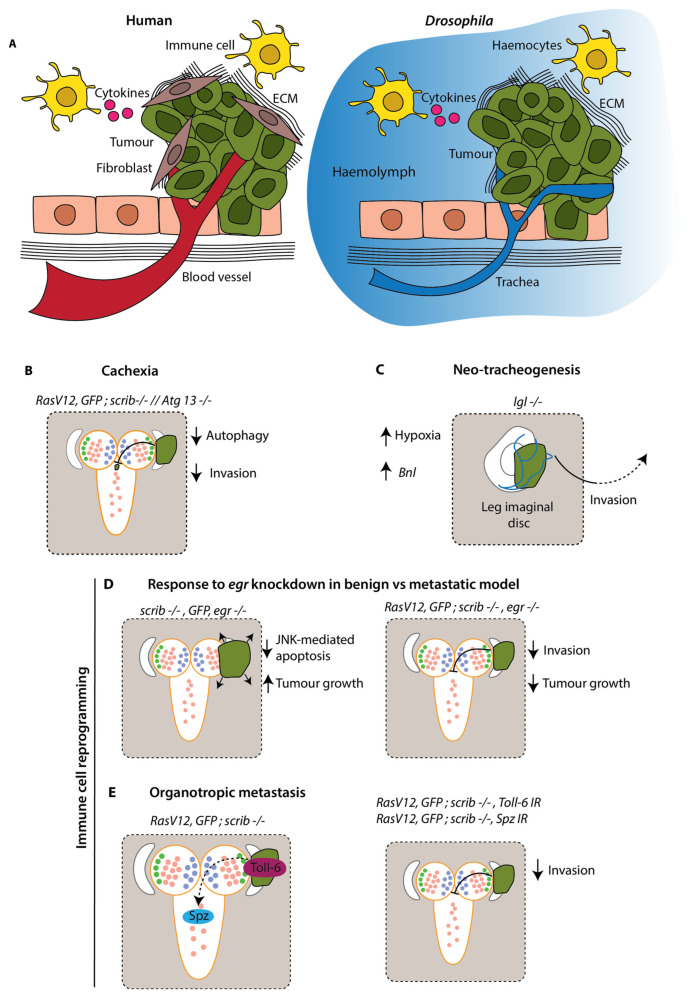
Modelling the influence of the tumour microenvironment in *Drosophila* larvae. (**A**) Key aspects of the tumour microenvironment in humans vs. *Drosophila*. The microenvironment includes fibroblasts, which secrete ECM and other factors to drive tumour growth and metastasis; immune cells, which can secrete cytokines and create an inflammatory environment; and blood vessels, which are recruited by the tumour and provide oxygen and nutrients to promote growth. *Drosophila* lack a closed circulatory system; their tissues are bathed in haemolymph and trachea supply oxygen. In the absence of an adaptive immune response and a vast array of immune cells, *Drosophila* rely on innate immune responses. In epithelia, wounds activate JAK/STAT and JNK signalling pathways which drives the recruitment of haemocytes. Haemocytes engulf and encapsulate foreign particles and initiate an inflammatory response. (**B**) Cachexia, where tumours cause wasting of healthy tissue, is a driving force of tumour growth. A reduction in autophagy protein Atg13 in surrounding tissue prevents the recycling of nutrients for the growing metabolic demands of tumours and reduces invasion [70]. (**C**) Hypoxia is a common feature of tumours, where their intense metabolic demands result in an inadequate oxygen supply to the tissue. Increased hypoxia in the tumour microenvironment results in recruitment of trachea via increased *Branchless* (*Bnl*) expression and promotes invasion [65]. (**D**) In a benign *scrib −/−* tumour, JNK signalling induces apoptosis and reduces tumour growth. When *Drosophila* TNFα (*egr*) is knocked down, loss of apoptosis drives tumour growth. In metastatic tumours, RasV12 confers an insensitivity to apoptosis and harnesses invasive capabilities of JNK signalling. Therefore, knockdown of *egr* in a *RasV12*; *scrib −/−* model inhibits invasion [58]. (**E**) Spätzle, a Toll-6 receptor ligand, is secreted at metastasis-receptive sites and engages JNK signalling by binding Toll-6 receptors on tumour cells, thus resulting in preferential colonisation of these sites. Preventing this interaction, either by knockdown of Toll-6 or Spätzle, results in reduced invasion into the VNC [60].

It is becoming increasingly apparent that the tumour microenvironment plays a key role in tumour growth, invasion and metastasis. This is in part thanks to work modelling cancer in *Drosophila* larvae. The accessibility and ease with which we can manipulate the microenvironment in this system has enabled phenomena such as neo-tracheogenesis, cachexia and immune cell reprogramming to be studied. However, despite the high number of mitotic cells in larvae making them amenable for modelling cancer, they remain as larvae for only 4–5 days, limiting the time that the tumours can be left to grow. This means that any changes in expression patterns or cell behaviour that occur after 5 days may be missed. One way to circumvent this problem is by using transplantation experiments.

## 5. Transplantation Experiments

Tumour allograft assays, or transplantation assays, involve the transplantation of tumour cells or tissue into another individual. The earliest reports of transplantation in *Drosophila* were at the start of the 20th century. In 1918, Mary Stark described “dark bodies” that resembled tumours in *Drosophila* larvae. In this pioneering work, Stark surgically removed these dark bodies and transferred them to healthy larvae in an attempt to examine the potential for these tumours to spread and cause host death [72]. Although inconclusive due to lethality associated with the surgery itself, this work laid the foundation for the next century of research using this technique. In 1936, a simple microinjection apparatus was developed, enabling successful transplantation between larvae without lethality [73]. At this time, cancer research in flies was in its infancy, and it was not until the 1960s that transplantation experiments were used again to investigate cancer in *Drosophila* [74]. After discovering that mutations in the tumour suppressor *lgl* led to the growth of invasive and lethal tumours in the larval brain and imaginal discs, Gateff and Schneiderman used a serial transplantation technique [75] to demonstrate that *lgl* tumour cells can survive multiple transplantations and metastasise in the adult host [74].

Early transplantation experiments also assisted in distinguishing between malignant and benign neoplasms in *Drosophila.* Malignant neoplasms are categorised by rapid growth, invasion into adjacent tissue, metastasis, loss of structure and function and lethal autonomous growth after transplantations. In contrast, benign neoplasms (also known as hyperplasia) retain structure and function, are non-invasive and do not grow after transplantation into a new wild-type host. Therefore, mutant lines were checked for the presence of malignant tumours against these criteria, including by serially transplanting tumours into adult hosts and examining their growth and histological characteristics in situ and after transplantation [31]. However, the technique fell into disuse and practically disappeared towards the end of the 20th century, only a few groups were aware of its potential and used it during this time. One such study by Woodhouse et al. demonstrated that imaginal disc tissue from larvae carrying tumorigenic mutations were in fact metastatic when transplanted into adult hosts, in contrast to in situ where they do not metastasise [76]. More recently, transplant experiments have become a standard method to analyse metastatic potential in adult flies. The revival and growth of the transplant technique in *Drosophila* cancer research was aided by the development of a standardised protocol specifically for studying tumour growth in *Drosophila* using the tissue allograft method [77].

A key advantage of the technique is enabling tumour growth to be monitored beyond the relatively short lifespan of a single larva. Researchers have made use of this to study the potential for metastasis of tumours generated in mitotic cells within the larvae in an adult host. The first study interrogating mutations sufficient to stimulate invasion and metastasis used transplantation experiments to confirm metastatic ability in the adult after observing metastasis in larvae (Figure 4A) [39]. For example, transplantation of the *Ras^V12^*; *scrib^–/–^* imaginal tumours discussed in the previous section resulted in rare metastasis and invasion of adult host tissues, including the ovaries and gut [39].

Serial transplants, where tumours are repeatedly harvested and retransplanted into new adult hosts, have allowed the study of primary tumour growth and metastasis over an even longer period (Figure 4B). Caussinus and Gonzalez (2005) demonstrated that tumours generated from larval neuroblasts carrying mutations in genes that control asymmetric cell division could grow to 100 times their initial size, invade other tissues and kill adult hosts within two weeks. These tumours have been serially transplanted for over two years and continue to grow, indicating that these cells could proliferate without end, unlike wild-type imaginal disc cells that could survive for years but do not proliferate [78]. Furthermore, small tumour colonies were found distal to the transplant site, suggesting that these tumours with perturbations to asymmetric cell division could metastasise in the adult host [78].

Whilst the presence of secondary tumours far from the transplant sites in adult hosts is highly suggestive of metastatic behaviour, it is possible that it is an artefact from injection into an open circulatory system. As the dissected tissue and fluid in the syringe is forced into the abdomen, the tumour may break up and travel passively, carried by the flow of injected fluid and haemolymph, to distant sites. This could result in tumour fragments appearing as metastases having not undergone the complex cellular transitions required to disseminate from the primary tumour and subsequently arrest and recolonise a secondary site. To address this problem in an existing transplant model, Beaucher et al. developed an in vivo assay for the metastatic potential of tumour cells by quantifying micrometastasis formation by immunofluorescence within the ovarioles of adult hosts after transplantation into the abdomen. In order to be found within the ovarioles, the tumour cells must actively pass through basement membranes and multiple cell layers (Figure 4C) [41]. This study built on prior work demonstrating that mutations in the tumour suppressor genes *lgl* and *brat* were sufficient to drive metastasis in the adult host [40,76]. Briefly, loss-of-function mutations in these genes trigger neoplastic overgrowth in brains and imaginal discs [39,76,79,80]. When transplanted into the abdomen of adult hosts, brain tumour fragments from *lgl*, *dlg* or *brat* mutant larvae were subsequently found in distant sites such as the leg, wing and head [40,76]. By examining the ovarioles for the presence of tumour cells, Beaucher et al. were able to confirm whether cancer cells were able to actively disseminate and colonise new sites [41]. This critical evaluation of the metastatic ability of *lgl* and *brat* tumours revealed that both were capable of the complex set of cell behaviours required for spread to the ovarioles.

Previously, Beaucher et al. demonstrated that whereas *lgl* and *brat* tumours had a similar rate of metastasis in the first instance, continuous passaging of the tumour cells into new hosts increased the rate of metastasis in *lgl* but not *brat* mutants [40]. Furthermore, non-invasive primary *brat* and *lgl* tumours contained cells expressing either neuronal (ELAV) or glial (REPO) markers but never both. In contrast, almost all *lgl* micrometastases expressed both markers. In *brat* secondary tumours, it was more variable, with less than half of the micrometastases expressing either marker. Using their newly developed assay for metastasis in the ovaries, Beaucher et al. were able to explore the mechanisms underlying these differences. They found that the matrix metalloproteinase MMP1 was required for colonisation of the ovarioles in both tumours, but the source of MMP1 was different. *lgl* tumours express MMP1 themselves, whereas *brat* tumours rely on increased *Mmp1* expression in the ovaries for metastasis [41]. This highlights the importance of tumour–microenvironment interactions in determining metastatic potential and is an example of how *Drosophila* transplant models can be utilised to investigate this.

In recent years, several studies have used tumour allograft assays to link metastasis to mutations in Notch signalling, inflammation and TGF-β signalling, among other pathways [39,78,81,82,83]. There is no doubt that transplant experiments have been instrumental in our understanding of tumour growth and invasion. However, the contribution of tumour microenvironment is relatively unexplored using this technique, despite its potential. One study harnessed the amenability of the transplant technique to confirm findings from larvae showed that autophagy in cells surrounding the *Ras^V12^*; *scrib* tumours is necessary for invasion from the eye disc into the VNC [70]. The *Ras^V12^*; *scrib*, *Atg13* tumours, which had limited growth due to ablation of autophagy, remained small when transplanted into autophagy-deficient hosts but proliferated when transplanted into wild-type hosts, thus demonstrating the importance of autophagy and the microenvironment in tumour growth [70]. Manipulating the microenvironment through the host genotype is an easily accessible but relatively under-utilised method to explore the relationship between specific tumour mutations and environmental conditions and the effect this has on metastatic potential.

Overall, transplantation experiments have advantages over studying metastasis in situ in larvae. The main advantage of this approach is the ability to overcome the limitation of the short larval lifespan, enabling tumour growth to be monitored for longer and the effect on adult lifespan to be examined. Tumours can be aged far beyond the lifespan of a single fly using serial transplantations, allowing changes in metastatic potential to be measured over a longer period, more relevant to the human tumour lifespan. Additionally, serial transplantations overcome an inherent technical challenge with *Drosophila* when aiming to collect large amounts tissue for omics and sequencing approaches, in that flies are very small. By continually harvesting and transplanting the tumours, sufficient tissue can be generated for these experiments.

Despite these advantages, there are drawbacks to this technique which should be considered. Firstly, one could argue that it is quite divorced from how cancer actually occurs, and it is important to consider this when using it for modelling purposes. Although the transplant technique has been made more accessible since the publication of a standard protocol [77], and more recently an automated method for injection [84], it is still time consuming and challenging. The labour-intensive nature of the approach prevents its use for large- or medium-scale genetic and drug screens. Additionally, it is important to properly define metastases by looking for them inside tissues that are surrounded by basement membrane, such as the ovaries [41], or by fluorescently labelling the basement membrane [39], otherwise secondary tumours could be an artefact from injection. Another major limitation of this technique is that it misses the first stages of tumour development. Transplant experiments focus on the later stages of metastases once the primary tumour is well developed, so we may be missing key events in initiation that happen early in primary tumour formation. Furthermore, serially transplanting tumours into new hosts allows the tumours to evolve, whereas the microenvironment is continually replaced. This is different to in human cancer where the tumour and microenvironment are adapting and responding to each other simultaneously. In summary, although transplant experiments have proved instrumental in developing our understanding of factors driving metastasis, there are practical disadvantages and inherent limitations associated with the technique.

## 6. Inducing Metastatic Tumours in Adult *Drosophila*

The requirement for a cell to divide for neoplastic transformations to be generated has limited the use of adult *Drosophila* for studies of cancer progression and metastasis. For a long time, the blood cells and the gonads were thought to be the only cells in adult *Drosophila* which undergo cell divisions. However, the discovery that the adult midgut is under constant renewal, with intestinal stem cells constantly dividing to replenish the tissue, opened a whole new system for cancer modelling in adult flies (Figure 4) [85,86]. This discovery allowed for new models of tumorigenesis based on stem cells in an adult organ that is remarkably similar to its vertebrate equivalent [87]. Furthermore, mutations in genes commonly found mutated in human colorectal cancer (CRC) were demonstrated to also lead to the formation of tumours in the adult fly intestine [88,89].

*Ras* is one example of a gene frequently mutated in human cancers [90]. Following a similar pattern to the development of human cancers, mutations in *Ras* alone are not sufficient to cause tumours in the *Drosophila* adult gut or larval imaginal tissues but instead lead to an over-proliferation phenotype (hyperplasia) [88,91]. However, as discussed in detail earlier, *Ras* mutations can act cooperatively with mutations in other tumour suppressors or oncogenes, for example the tumour suppressors *scrib* [42] or *Adenomatous polyposis coli* (*APC*) [88], to induce the growth of benign tumours.

The *APC* gene is found mutated in 60% to 75% [89,92] of human CRCs. *APC* encodes a protein that inhibits Wnt pathway activation and is named for the thousands of adenomatous polyps found in the gut of patients with familial *APC* mutations, at least one of which has an almost 100% chance of becoming cancerous [93]. In *Drosophila*, loss-of-function mutations in both the *Apc* genes in combination with oncogenic *Ras* mutations leads to the formation of large tumours that grow aggressively, either inwards towards the lumen of the gut or outwards towards the surrounding musculature [88,94].

Although these models recapitulate various aspects of human cancers, the resulting tumours are constrained by the ECM and do not metastasise. It is important to remember that the process of metastasis involves invasion and detachment of tumour cells into the circulatory system, transport around the organism, arrest at a suitable location, extravasation into the surrounding tissue and proliferation into a viable metastasis [95]. As previously mentioned, flies have an open circulatory system and no adaptive immune response, thus the process of metastasis in mammals cannot be perfectly replicated in *Drosophila*. However, a number of models where one or more steps of metastasis are recapitulated in adult flies have now been developed.

The first model showing dissemination of mutant cells in adult flies used overexpression of the oncogenic allele *Ras^V12^* to induce normally quiescent stem cells in the *Drosophila* hindgut to proliferate [96]. These cells can disrupt the basal lamina to invade out of the hindgut and can be found individually or in clusters at distant sites within the fly (Figure 5B). This process can be enhanced by causing a sustained immune response in the gut through bacterial infection [96]. Lee et al. demonstrated a similar pattern of dissemination in the midgut by overexpressing *Ras^V12^* in all ISCs [97]. This dissemination requires the metalloprotease MMP1 to break the basement membrane, downstream of the mechano-sensor Piezo [97]. These models provide important insights into the factors involved in dissemination of tumour cells; however, as overexpression of oncogenic *Ras^V12^* alone does not cause the development of primary tumours in the first instance, they are missing key aspects of the metastatic cascade.

Tumours can also be induced in the developing pupal eye by disrupting the Notch signalling pathway. Notch signalling is required for normal cell proliferation and differentiation in the *Drosophila* midgut. It has also been found to be disrupted in a number of human tumours [99]. Disruption of Notch signalling, through either suppression of Notch in daughter cells [86] or reduced levels of Notch ligand Delta in the ISCs [100], leads to the growth of tumours enriched in ISCs and secretory enteroendocrine (EE) cells. In this study, overexpression of Delta leads to a large eye phenotype; however, if Delta is overexpressed alongside loss-of-function mutations of the axon guidance regulator *lola* and *psq*, a gene involved in retinal cell fate determination, large tumours develop in the eye [101]. These tumours form large secondary metastases in 30% of adult flies. In this model, tumour growth can be prevented by restoring expression of Retinoblastoma-family protein (Rbf), a *Drosophila* homologue of the retinoblastoma family of tumour suppressors, which was found to be hypermethylated. This gives an opportunity to study the effects of the interaction between genetics and epigenetics in the generation of primary and secondary tumours.

A more complete model of the metastatic cascade in adult *Drosophila* was generated in 2019, building on the aforementioned work by Martorell et al. in which a model carrying null mutations in endogenous *Apc* genes and overexpressing oncogenic *Ras^V12^* exhibited growth of tumours in the fly midgut [88]. Constrained by the ECM, these tumours did not invade surrounding tissue or metastasise. However, driving tumour cells to undergo an EMT through overexpression of the EMT transcription factor Snail in *Apc*, *Ras^V12^* flies, led to the formation of tumours capable of breaking through the basal lamina, migrating collectively and forming large metastases in the abdomen, thorax and head (Figure 5C) [98]. Activation of EMT has been implicated in several human cancers [102], and this work in *Drosophila* also mirrors research in mice, where a reduction in levels of EMT transcription factors have been shown to reduce the number of metastases [103], whereas increased levels of EMT transcription factors correlate with the number of metastases [104]. Understanding more about the role of EMT in metastasis and how we could target this therapeutically is an important focus for future research. The development of a high-throughput screening technique measuring circulating tumour cells and whole tumour burden using luciferase activity makes this model amenable to both genetic and drug screening [105].

As with transplant assays, adult models of metastasis allow the development of a tumour to be studied over a longer period, better recapitulating human disease. In contrast to transplant experiments, generating clones in situ using MARCM in adult flies circumvents practical issues with microinjection, making such models far more amenable to high-throughput screening. In addition, a secondary tumour forming in the adult tumour model cannot be an artefact of the injection and importantly, it allows many stages of metastasis, including the very first and last, to be followed in a single adult fly.

## 7. Conclusions and Future Directions

Since the earliest discoveries of cancer-related genes in larvae, the development of rudimentary transplantation experiments demonstrating the differences between benign and malignant neoplasms and, more recently, the development of adult tumour models, research using fruit flies has been instrumental in enhancing our understanding of factors underlying cancer. An important focus of future cancer research is the metastatic cascade. Metastasis is the biggest cause of cancer-related death, but the mechanisms remain unclear. *Drosophila* models are ideally placed to address this, and they have already yielded significant findings implicating EMT, dysregulated cell signalling and different aspects of the microenvironment in metastasis.

Each type of *Drosophila* metastasis model has advantages and disadvantages, and each is ideally suited to investigating distinct aspects of metastasis using different techniques. Although larvae have large populations of mitotic cells amenable to induction of tumour growth, they are limited in the time the tumours can grow and are thus unsuitable for long term studies of tumour behaviour. This can be overcome by transplanting larval tumours into adult hosts and monitoring their growth. The main disadvantage of this method is that the primary tumour did not grow in the adult host, meaning you are missing the initial key steps in the metastatic cascade, as well as inducing possible artefacts through injection. More recently, adult models have been developed based on the finding of mitotic intestinal stem cells in the adult gut. These models are advantageous because you can model the entire metastatic cascade and monitor growth and behaviour over long periods. However, all *Drosophila* models are based on the expression of markers such as GFP, RFP and luciferase. It is possible that some of these markers may affect cell behaviour, survival and response to the immune system. Therefore, it may be an important future endeavour to establish endogenous markers of tumorigenesis to study how tumours behave in the absence of these ectopic factors.

In summary, as a model for metastasis *Drosophila* has limitations but offers also important advantages that can be exploited. There are now a great variety of metastasis models, from the genetically induced to transplantation experiments, that are continuing to contribute to our understanding of how cells become metastatic. Moreover, the great amenability to genetic manipulation that *Drosophila* offers will help to dissect how gene expression is able to modulate the extraordinary cell plasticity shown by tumour cells and how it is required for their adaptation to new microenvironments.

## Figures and Tables

**Figure 1 cells-12-00677-f001:**
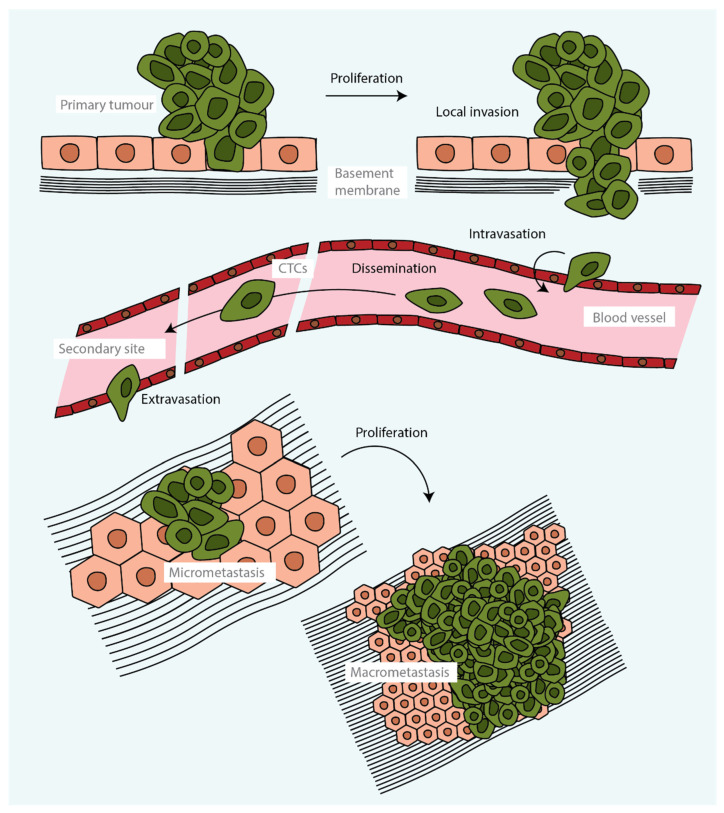
The metastatic cascade. Following local invasion of the basement membrane at the primary tumour site, cells can enter the blood vessel or lymphatic system and disseminate. Circulating tumour cells (CTCs) travel alone or in clusters and eventually pass through the endothelium and infiltrate distant organs. Cells may remain dormant as micrometastases (typically 2–100 cells) or proliferate and form a secondary tumour.

**Figure 2 cells-12-00677-f002:**
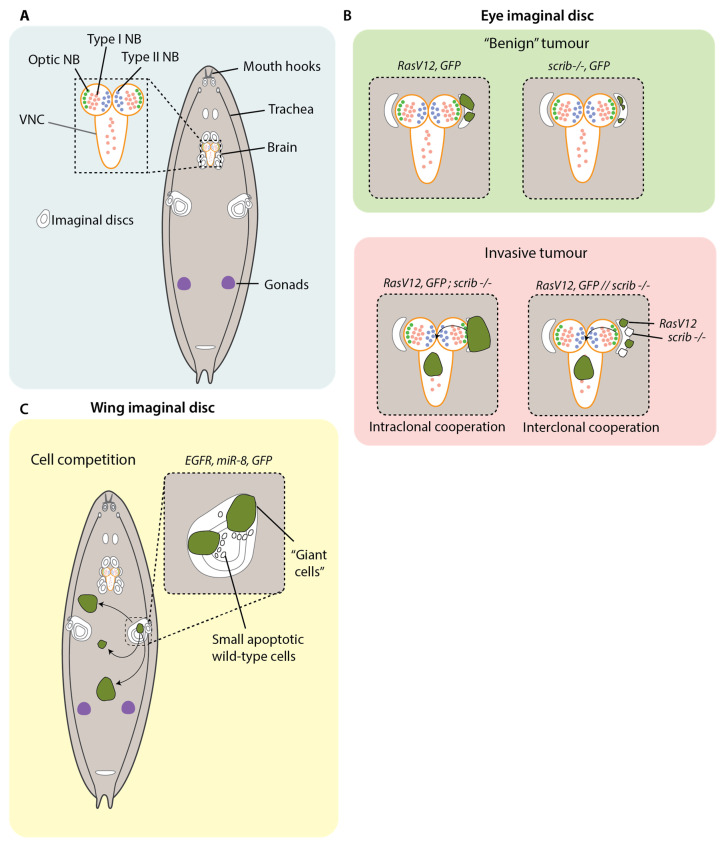
Modelling metastasis in *Drosophila* larvae. (**A**) *Drosophila* larva, showing tissues that undergo mitosis: neuroblasts, imaginal disc cells and gonads. NB = neuroblast; VNC = ventral nerve cord. (**B**) *Ras^V12^* and *scrib−/−* tumours in the eye imaginal disc form non-invasive tumours. Intraclonal (both mutations within the same clone) and interclonal (mutations in adjacent clones) cooperation between of loss of a tumour suppressor gene such as *scrib* and expression of an oncogene such as *Ras^V12^* results in invasion of the VNC that is reminiscent of the initial stages of metastasis [32,33]. Intraclonal cooperation has also been observed between *Ras^V12^* and mutant mitochondrial respiration complexes [34]. (**C**) Tumours over expressing *EGFR* and *mIR-8* in the wing imaginal disc grow into neoplasms that can disseminate throughout the larva. A subset of these cells develops into “giant cells” flanked by differentiated wild-type cells. The smaller wild-type cells are engulfed by the giant cells in an apoptosis-dependent manner [35].

**Figure 4 cells-12-00677-f004:**
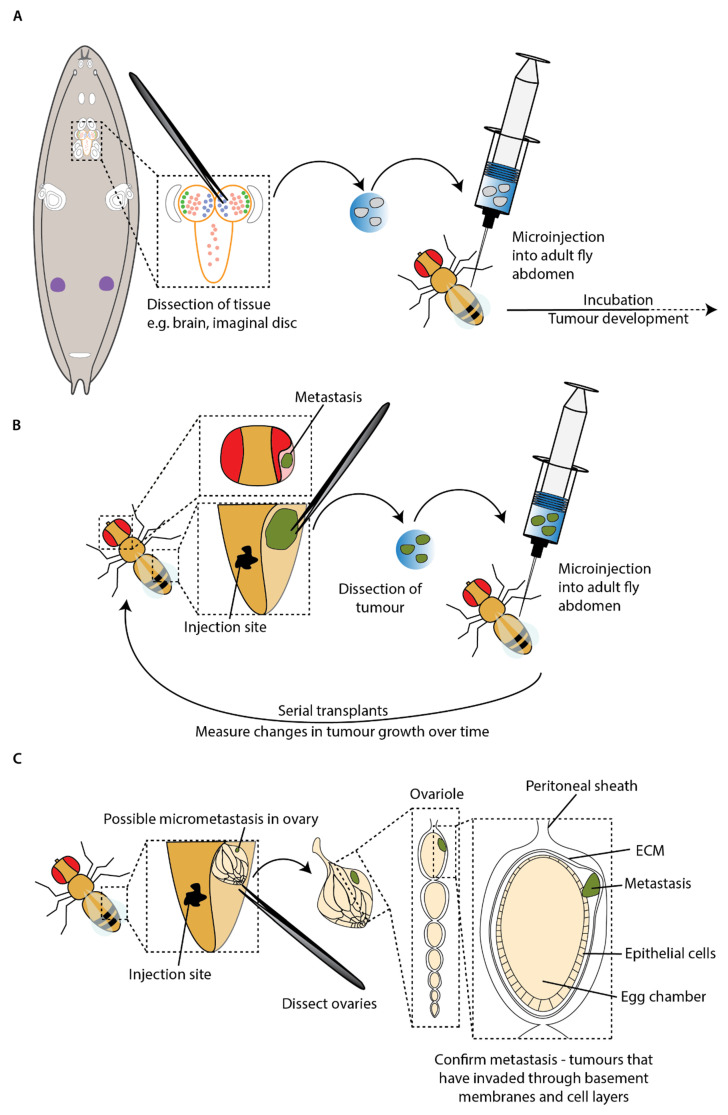
Transplantation experiments to model metastasis. (**A**) Transplantation of larval tissue into adult hosts. Tumours are induced in larvae, usually in the imaginal discs or the brain, and dissected into PBS. This is microinjected into the abdomen of adult host. (**B**) Serial transplants—tumour dissected from adult host and re-transplanted into a new host, allowing tumours to be incubated for several months. (**C**) Critical evaluation of active cell spread and colonisation of distant sites. Tumour masses could travel passively, carried by the flow of injected fluid and haemolymph, and be found distant to the transplant site. To be found in the ovary, tumour cells must pass through cell layers and basement membrane. This ensures that secondary tumours are not an artefact from injection into an open circulatory system [41].

**Figure 5 cells-12-00677-f005:**
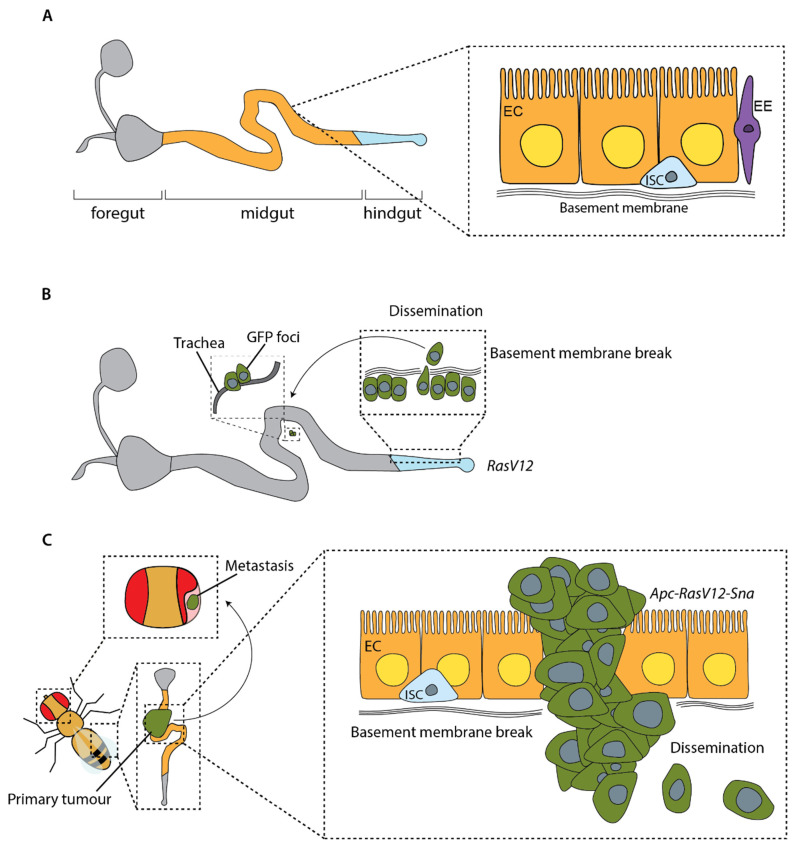
Modelling metastasis in adult flies. (**A**) The adult midgut contains pools of intestinal stem cells (ISCs) under constant renewal, which has enabled the development of adult cancer models. EC = enterocyte; EE = enteroendocrine cell. (**B**) Cells expressing *Ras^V12^* in enterocytes and their progenitors in the hindgut can disrupt the basal lamina to invade out of the hindgut and can be found individually or in clusters in the midgut and associated with trachea [96]. (**C**) Overexpression of the EMT transcription factor Snail in *Apc-Ras^V12^* flies enables tumour cells to break through the basement membrane, disseminate and form secondary tumours. These can be seen in the thorax and the head [98].

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
