# Peer review of "Modelling Cancer Metastasis in Drosophila melanogaster"

_cells, 2023, doi:10.3390/cells12050677_

Round 1

Reviewer 1 Report

This is a terrific manuscript.  I quite enjoyed the historical style, which I think will be valuable to others as well – both experts in the fields of Drosophila biology and cancer, and newcomers to both.  I have no comments, other than to urge the authors to revisit the standard nomenclature for Drosophila gene names.  There are a few places where gene names should be italicized, or not-capitalized, etc.

Four other extremely-minor points:

page 1, line 17, I’m not sure what “morphologically” is doing in the sentence, since phenotypes need not have morphological manifestations.

page 3, line 78, the sentence is confusing because it seems to be saying that flies have a lack of an open circulatory system.  A simple tweak of this sentence is sufficient.

page 6, line 163-164, the “dominant negative” should be desrcribed. Is this an antimorph or a toxic hypermorph?  Something about the nature of the lesion should be included.

page 6, line 201, I understand what “Mito” is meant to mean, but it seems that the authors are talking about a gene named “Mito.”  Some other way of describing this should be used.

Other than those, really great work. 

Author Response

We thank the reviewer to drive our attention to the minor points that have been corrected in the revised version.

Reviewer 2 Report

This is a well-written, comprehensive review of the field. Includes a historic perspective and contribution of the recent publications. Nice work!

Author Response

We thank the reviewer for the support to the publication of our work.

Reviewer 3 Report

The review by Sharpe et al., entitled “Modelling cancer metastasis in Drosophila melanogaster” focus of the use of Drosophila as a model for cancer metastasis.

The article is very well written and documented, with a history of the field. The figures help to understand the text in a pleasant way. Nothing more to add for me. 

Author Response

(The authors gave the same response as above.)

Reviewer 4 Report

The  paper from Dr. Sharpe and coworkers is a thorough review article on the use of Drosophila as a model for cancer metastasis.

Overall this manuscript is very well written with very explicative figures that facilitate the reader.  

I suggest a few changes:

-The manuscripts lacks of an abstract or an introduction.Maybe the first paragraph entitled Introduction should correspond to an abstract.

-The figure legends are sometimes succinct: I suggest to describe the panels in the figures more accurately.

-The authors could add a new figure to illustrate the use of Drosophila larvae to study the interactions with the microenvironment.

-The conclusions should include a section/paragraph in which the authors can discuss the advantages and the limits of the Drosophila with respect to other animal models.

Author Response

We thank the reviewer for the comments on our review. Following the suggestions we have added and abstract, added a new figure (Figure 3) to illustrate the use of Drosophila larvae to study the interactions with the microenvironment and updated and expanded the figure legends.

Regarding the comment about a section/paragraph in which we could discuss the advantages and the limits of Drosophila with respect to other animal models, we think that this is already reflected throughout the text, and a proper comparison between different models would require another review in itself.